# Analysis of the Level of Plasmid-Derived mRNA in the Presence of Residual Plasmid DNA by Two-Step Quantitative RT-PCR

**DOI:** 10.3390/mps3020040

**Published:** 2020-05-23

**Authors:** Barbara Ahlemeyer, Claudia Colasante, Eveline Baumgart-Vogt

**Affiliations:** Institute for Anatomy and Cell Biology, Division of Medical Cell Biology, Justus Liebig University, Aulweg 123, 35385 Giessen, Germany; Claudia.Colasante@anatomie.med.uni-giessen.de

**Keywords:** endogenous genome-derived mRNA, genomic DNA, DNA contamination, nonsense-tail reverse transcription, nonsense-tail PCR primer, overexpression experiments, plasmid DNA, plasmid-derived mRNA

## Abstract

In transfection experiments with mammalian cells aiming to overexpress a specific protein, it is often necessary to correctly quantify the level of the recombinant and the corresponding endogenous mRNA. In our case, mouse calvarial osteoblasts were transfected with a vector containing the complete *Pex11β* cDNA (plasmid DNA). The *Pex11β* mRNA level, as calculated using the RT-qPCR product, was unrealistically higher (>1000-fold) in transfected compared to non-transfected cells, and we assumed that there were large amounts of contaminating plasmid DNA in the RNA sample. Thus, we searched for a simple way to distinguish between plasmid-derived mRNA, endogenous genome-derived mRNA and plasmid DNA, with minimal changes to standard RT-PCR techniques. We succeeded by performing a plasmid mRNA-specific reverse transcription, and the plasmid cDNA was additionally tagged with a nonsense tail. A subsequent standard qPCR was conducted using appropriate PCR primers annealing to the plasmid cDNA and to the nonsense tail. Using this method, we were able to determine the specific amount of mRNA derived from the transfected plasmid DNA in comparison to the endogenous genome-derived mRNA, and thus the transfection and transcription efficiency.

## 1. Introduction

Reverse transcriptase quantitative PCR (RT-qPCR)-based gene expression analysis of cells transfected with plasmid DNA requires the isolated RNA to be free from contaminating genomic and plasmid DNA. Contaminant DNA might be co-amplified together with the cDNA, leading to background and false positive results, which is, for example, a problem in metagenomics analyses [1]. To determine the amount of contaminating genomic DNA in RNA samples, several PCR-based approaches have been proposed, e.g., the addition of a genomic DNA reference sample [2] and the detection of the intron/exon ratio of a housekeeping gene [3] and of ribosomal DNA [4].

There are two ways to avoid this issue in RT-qPCR. One possibility is to get rid of the residual DNA by choosing an appropriate RNA isolation procedure or by treating the RNA sample with DNase. However, in transfection experiments where the cell lysate contains a bulk of plasmid DNA, the standard RNA isolation and DNase treatments might give unsatisfactory results, since even very few copies of residual DNA molecules might lead to false positive results. This is especially true in cases when the analyzed mRNAs have a low abundance. A second possibility to avoid the amplification of contaminating DNA is to take advantage of the structural differences between mRNA and DNA. For example, the cDNA does not contain introns and is thus selectively amplified when exon-spanning primers are used during the qPCR reaction [5]. Alternatively, an oligo(dT) RT primer tagged with a unique sequence [6] can be used for reverse transcription, and the detection of the tagged cDNA during qPCR is conducted by using a reverse PCR primer matching the unique sequence. Both ways, however, will not work in the presence of processed pseudogenes or high amounts of residual plasmid DNA (e.g., in transfection experiments), as these DNA species do not have introns and poly(A) tails. To our knowledge, until now, only two research groups have dealt with the problem of DNA amplification in RT-PCR, but both are exclusively focused on avoiding pseudogene amplification, which is common for the reference genes *Gapdh* and *Actb* [7,8]. Hurteau and Spivack [7] used a poly(T)_21_ RT primer with a variable three nucleotides (nt) overhang at the 3′-end (thus binding to any mRNA) and a unique sequence of 18 nt at the 5’-end. Because the genomic DNA did not incorporate the unique tag sequence, it was suggested to be sufficient to obtain gene-specific (and no pseudogene) amplicons. Additionally, Smith et al. [8] used a primer with a variable 2 nt overhang and a poly(T)_18_ tail, tagged with a unique sequence of 24 nt, for reverse transcription, but in this study, the specificity of the *Gapdh* mRNA was enhanced by several rounds of step-out and step-in PCRs with limiting primer concentrations used for the first PCR rounds. Among the six resulting PCR products, the one that reflected the *Gapdh* mRNA level was distinguishable from the others by gel electrophoresis due to its size and detectable amount [8]. For qRT-PCR, this method is not applicable, because it generates multiple PCR products. In addition, genome-derived endogenous mRNA will be detected as well. Thus, we searched for a simple way, with minimal change to our current RT-qPCR protocol, to quantify the plasmid-derived mRNA level, without detecting residual plasmid DNA and endogenous genomic DNA and mRNA. We succeeded by modifying the primer design suggested by Smith et al. [8] and produced an RT primer, in which the first part annealed to the 3´-end of the plasmid-derived mRNA (as used in a step-in round PCR by Smith et al. [8]), followed by a poly(T)_17_ tail (which annealed to the poly(A)_17_ of the plasmid-derived mRNA) and, finally, a nonsense nucleotide sequence (as used by [6,7,8]). The qPCR was performed with PCR primers annealing to the 3´-end of the plasmid cDNA (forward primer) and to the nonsense sequence tag (reverse primer), which was incorporated into the plasmid cDNA during the RT-PCR. This allowed for the selective amplification of plasmid-derived cDNA and the quantification of the plasmid-derived mRNA level, even in the presence of contaminating plasmid DNA.

## 2. Materials and Methods

### 2.1. Materials

The mouse calvarial cell line (MC3T3-E1) was from the Deutsche Sammlung von Mikroorganismen und Zellkulturen (DSMZ, Braunschweig, Germany) and was established by Sudo et al. [9]. The mouse *Pex11β* cDNA expression pCMV-SPORT6 vector (IRAK p961C0923Q2) was obtained from the Deutsches Ressourcenzentrum für Genomforschung (RZPD), Heidelberg, Germany. The open reading frame of the *Pex11β* plasmid coded for nucleotides 258 to 1788 of the mouse *Pex11β* transcript variant 1 (IMAGE clone 3964491), representing exons 1–4 (no 5′-UTR) and half of the 3′-UTR (the first 753 nucleotides of the 3′-UTR), together with a poly (A)_17_ tail. The empty (backbone) pCMV-SPORT6 vector, MEM alpha medium (Cat. No. 22571-020), Opti-MEM™ I reduced serum medium (Cat. No. 31985070), High-capacity cDNA reverse transcription kit (Cat. No. 4368813), DNase I kit, Amplification grade (Cat. 18068015) and Maxima SYBR Green PCR kit (Cat. No. K0243) were purchased from Thermo Fisher Scientific (Dreieich, Germany). RNazol^®^ RT (Cat. No. R4533) was bought from Sigma-Aldrich Chemie GmbH (München, Germany). *Trans*IT^®^ LT-1 transfection Reagent (Cat. No. MIR2300) was derived from Mirus (through VWR, Darmstadt, Germany). The primers used for nonsense-tail reverse transcription and qPCR were synthesized by Eurofins (Ebersberg, Germany). The sequences and efficiency coefficients of all the primers used in this study are given in Table 1. The nomenclature of the mouse genes and proteins follow the guidelines of the official NIH nomenclature throughout the manuscript.

### 2.2. Transfection of the MC3T3-E1 Cell Line

In osteoblast-like MC3T3-E1 cells, peroxisomes and especially the peroxisome biogenesis protein *PEX11β* was thought to be essential for osteoblastogenesis, lipid metabolism and redox balance [10,11,12]. To gain further insight into the function of *PEX11β*, we overexpressed this protein by transfecting the cells with the respective mouse cDNA-expression vector. For this purpose, cells were seeded into 12-well plates at a density of 10^4^ cells per cm^2^. Cells were cultured for 24 h in an MEM alpha medium, without antibiotics (1 mL/well), and were then either transfected with a plasmid containing mouse *Pex11β* cDNA, which was inserted into the pCMV-SPORT6 vector (*Pex11β* vector), with an empty pCMV-SPORT6 vector (empty vector), or with transfection reagent only (no transfection). The transfection was performed using the *Trans*IT^®^ LT-1 transfection reagent according to the manufacturer’s instructions. Specifically, 1 µg of plasmid DNA was added to 100 µL of an Opti-MEM™ I reduced-serum medium and gently pipetted to mix them together completely. Next, 3 µL of *Trans*IT^®^ LT1 reagent was added to the diluted DNA, gently pipetted and incubated at room temperature for 30 min. The transfection reagent:DNA complexes were added dropwise to different places of the 12-well area. Cells were harvested for RNA isolation 6 h and 12 h after transfection.

### 2.3. RNA Isolation, RT Reaction and Quantitative PCR Analysis

The total RNA was isolated using an RNazol^®^ RT reagent. In brief, cells were collected without being washed, in 1 mL of the RNazol^®^ RT reagent and were homogenized after adding 0.4 mL of water. The mixture was allowed to stand for 15 min and then centrifuged at 12,000× *g* for 15 min. The supernatant (containing the total RNA) was transferred to a new tube and mixed with an equal amount of 100% isopropanol to precipitate the RNA. The pellet was washed twice with 75% ethanol and solubilized in RNase-free water. The amount and quality of the total RNA was evaluated using a NanoDrop 2000 UV-VIS spectrophotometer (peqlab, Erlangen, Germany). Only RNA of a high quality, with A260/280 nm of >1.95 and A260/230 nm ratio of >1.8, was used. The RNA template was either used directly for subsequent RT reactions or treated with DNase I (0.3 U/µL of the reaction buffer).

First-strand cDNA was synthesized from 2 µg of the total RNA with either random hexamers (random RT) or nonsense-tail primers (nonsense-tail RT) or water (no primer RT) and dNTPs using the MultiScribe™ reverse transcriptase from the High- Capacity cDNA Reverse Transcription Kit. The MultiScribe™ reverse transcriptase resembles the MuLV reverse transcriptase [13], which selectively transcribes single-stranded RNA into the complementary cDNA strand with a low RNase H activity (this enzyme degrades RNA from DNA-RNA hybrids). In parallel experiments, the reaction was performed without reverse transcriptase to detect residual genomic DNA (Table 2).

For qPCR, we used the Maxima SYBR^®^ Green Mastermix, which was mixed with the template cDNA and the forward and reverse primers (1:1). All samples were run in triplicates in each series of experiments. The PCR reaction was conducted in the My iQ™2 iCycler (Bio-Rad Laboratories, München, Germany) using the following two-step amplification protocol: 10 min at 95 °C (enzyme activation), 40 cycles of 15 s at 95 °C (denaturation), and 60 s at 60 °C (annealing and extension, Table 2).

The specificity of the primer pairs for qPCR was evaluated, and they all showed a single peak in the melting curve analysis. The correct sizes of all amplicons are shown in a 3% agarose gel in Appendix A. Their amplification efficiency coefficient (E) for each primer pair (Table 1) was evaluated by 10-fold dilutions series using the slope of the regression between the log values and the average *ct* values:$$E=10^{\left(\left(-1\right)/slope\right))}-1$$

The PCR product levels between two experimental groups were compared through calculation using the 2^–ΔΔCT^ method of Pfaffl et al. [14]. For example, we compared the PCR product levels (relative expression ratio = R) of the cDNA of the empty vector- and *Pex11β* vector-transfected cells (defined as samples 1 and 2) using the following equation (Equation (1)).
(1)$$R=\frac{E1^{\left(ct\;sample\;2-ct\;sample\;1\right)for\;PCR\;primer\;1}}{E2^{\left(ct\;sample\;2-ct\;sample\;1\right)\;for\;PCR\;primer\;2}\;}$$
where *E* is the efficiency coefficient of the respective PCR primer (pair), e.g., E1 for primer 1 and E2 for primer 2, and *ct* is the number of cycles needed to reach a defined level of fluorescence, e.g., ct sample 2 for primer 1 means the *ct* value of sample 2 using the PCR primer pair 1.

In other reactions, we compared the PCR product levels of the same cDNA using different primers for the PCR reactions. In this case, normalization against a reference gene is not applicable. To compare the PCR product levels, we first adjusted the *ct* value of each primer to the value it would be in the case of an efficiency coefficient of 2 (Equation (2a)).
(2a)$$ct\;\left(adjusted\;to\;E=2\right)\;for\;primer\;n=1/\ln(En^{ct\;primer\;n})$$

The adjusted *ct* values were used to directly compare the PCR product levels obtained using the two different primer pairs through Equation (2b).
(2b)$$R=2^{\left(ct\;\left(adjusted\;to\;E=2\right)\;of\;primer\;2-ct\;\left(adjusted\;to\;E=2\right)\;of\;primer\;1\right)}$$

For all data discussed in the Results and Discussion chapter, we mention which experimental groups were compared to each other and which calculation method (either Equation (1) or Equation (2a) followed by Equation (2b) (Equation (2)) was used. The respective *ct* values are given in the text with reference to their position inside Table 3, Table 4 and Table 5.

## 3. Results and Discussion

### 3.1. Detection of Contaminating Plasmid DNA in Transfected MC3T3-E1 cells

The peroxisome biogenesis protein *PEX11β* is thought to be a key player in the regulation of peroxisome abundance [10,11,12], but the mechanism by which it affects peroxisome dynamics is still not fully understood. To obtain further insight into the function of *PEX11β*, we transfected MC3T3-E1 cells with the respective mouse cDNA-expression vector for overexpression studies. To survey the success of our transfection, we analyzed the endogenous genome- and plasmid-derived *Pex11β* mRNA levels by RT-qPCR using random hexamers for RT and *Pex11β* exon 3–4 spanning (termed 11b_Ex34) PCR primers for the following qPCR. The *Pex11β* mRNA level was normalized to the mRNA level of the reference gene, *Hprt*, which was detected using *Hprt* exon 1–2 spanning PCR primers (termed Hprt_Ex12). During this process, we measured extremely high increases in the 11b_Ex34 PCR product levels in the *Pex11β* vector-transfected compared to the empty vector- transfected cells, 6 h and 12 h post-transfection (Table 3, column 4, line 1 vs. line 2, and line 3 vs. line 4, respectively; calculation based on Equation (1)).

However, through closer investigation, the RT-qPCR product was found to be an artefact derived from amplified plasmid DNA (probably co-precipitated during the RNA isolation procedure; Figure 1, arrow in the middle from line 4 to line 6). We assumed that the amplicon was not derived from the genomic *Pex11β* DNA, because the intron between exon 3 and 4 was 6237 nt long, and due to its size, it cannot be amplified in qPCR. Indeed, in the absence of RT, no PCR product was found with the 11b_Ex34 PCR primer (Table 4, column 3, line 9; Figure 1, line 1 showing introns between exon 3 and 4 in the genomic *Pex11β* DNA). The same was true for the Hprt_Ex12 PCR primer, where no PCR product was found in non-transfected cells without a prior RT reaction (Table 3, column 2, line 5; Table 4, column 2, line 9). Next, we treated the RNA sample with DNase I, which, however, did not remove the residual plasmid DNA entirely (Table 3, column 4, line 4 vs. line 6).

We had already used a three-fold higher amount of DNase I, as recommended by the manufacturers of the DNase I kit and further increases of the DNase I concentration may lead to RNA degradation [15,16]. Thus, we tried to find a simple RT-qPCR protocol to distinguish between plasmid-derived mRNA, endogenous genome-derived mRNA and (residual) plasmid DNA.

### 3.2. A Simple RT-qPCR Protocol to Distinguish between Plasmid-Derived mRNA and Plasmid DNA/Endogenous Genome-Derived mRNA

We performed reverse transcription using a complementary RT primer consisting of the last 12 nucleotides of the *Pex11β* cDNA sequence, which were inserted into the plasmid vector, followed by a poly(T)_17_ and by a nonsense nucleotide sequence, with no similarities to the sequences of the mouse transcriptome. This RT primer was termed the nonsense-tail RT primer (Figure 1 in the top box on the right and lines 2B and 4B; Table 1). With regard to the plasmid-derived mRNA, the first part of this primer binds to the end of the short 3′-UTR, adjacent to the beginning of the poly(A) tail, to which the poly (T)_17_ part of this primer anneals (Figure 1, line 5B). In the mouse endogenous genome-derived mRNA, there is a gap of 651 nt between the annealing site of the first part and of the poly(T)_17_ part of this primer originating from the endogenous 3′-UTR length (Figure 1, line 2B, on the left). This should result in an inefficient annealing and reverse transcription of endogenous genome-derived mRNA and allow for a differentiation between genome- and plasmid-derived mRNA (Figure 1, line 2B on the left). In addition, the poly(T)_17_ part, which is in the middle of the RT primer sequence, will hardly bind to poly(A) tails longer than 17 nt, such as those found in endogenous genomic mRNA (Figure 1, line 2B on the right). The poly(A) tails of genome-derived mRNA are of variable lengths, but mostly contain 150–250 adenosine residues, and the shorter ones contain at least 50–100 nt [17].

The detection of the reverse transcribed plasmid-derived mRNA (plasmid cDNA), tagged with the nonsense-tail RT primer sequence, was conducted in the subsequent qPCR using a PCR primer pair binding upstream of the 3′-UTR region (forward primer) and to the nonsense tail with an overlap of poly(T)_4_ (reverse primer). This PCR primer pair was termed 11b_Tail (Figure 1, in the lower box on the right, Table 1). The results of our experiments are shown in Table 4.

The following conclusions can be drawn from these data:

(1) The nonsense-tail RT primer allowed for the efficient reverse transcription of plasmid-derived mRNA into plasmid cDNA (Figure 1, line 5B). This is shown by the high abundance of the 11b_Tail PCR product (ct = 13.2; Table 4, column 1, line 2) in *Pex11β* vector-transfected cells.

(2) We exclusively detected plasmid-derived mRNA and not plasmid DNA, as shown by the absence of an 11b_Tail PCR product in *Pex11β* vector-transfected cells, when random hexamers were employed during reverse transcription (Table 4, column 1, line 4).

(3) A negligible amount (0.7%) of the total genomic DNA-derived mRNA was detected when the nonsense-tail RT and 11b_Tail PCR primer PCR were employed. We came to this assumption, because we unexpectedly found a PCR product (ct = 24.6, Table 4, column 1, line 1) in empty vector-transfected cells. Where does this PCR product come from? Theoretically the nonsense-tail RT primer could non-specifically bind to endogenous genome-derived mRNAs, although to a negligible extent, compared to plasmid-derived mRNA. For example, the RT primer might bind, via its d(T)_17_ part, to any mRNA, or the 12 nt overhang may favor binding to endogenous genome-derived mRNA with complementary sequences. This would generate nonsense tail tagged endogenous mRNA-derived cDNA during reverse transcription, but the subsequent qPCR would be extremely inefficient, since it is very unlikely that the very specific forward primer of the 11b_Tail PCR primer pair will bind to this cDNA (Figure 1, line 2B on the right). Additionally, endogenous, genome-derived *Pex11β* mRNA would not be detected by qPCR, since the 11b_Tail PCR primer PCR would generate an excessively long amplicon (730 bps) (Figure 1, line 2B on the right). It is more likely that the RT primer will bind to the middle part of the 3′-UTR of the endogenous genome-derived mRNA via the 12 nt at the 3′-end, and this would add an overhang of d(T)_17_ and the 17 nonsense-tail nt to the cDNA (Figure 1, line 2B on the left). In this case, the resulting endogenous genome-derived mRNA-derived cDNA would be nearly identical to the plasmid-derived cDNA: it would be tagged with the nonsense-tail, but with a shorter 3′-UTR, and could be detected in the subsequent 11b_Tail PCR primer PCR (Figure 1, line 2B on the left). Thus, we assumed that the PCR product of the nonsense-tail RT and 11b_Tail PCR primer PCR of empty vector-transfected cells derives from this small amount (0.7%) of non-specifically amplified genomic DNA-derived mRNA. The percentage was calculated by comparing the ct values of the products of the 11b_Tail PCR (ct = 24.6 Table 4, column 1, line 1) and the 11b_Ex34 PCR (ct = 17.5, Table 4, column 3, line 1) based on Equation (2). The 11b_Ex34 PCR product was generated independently of the presence of the RT primers and thus reflects the total genomic DNA-derived mRNA present in the sample. The same PCR product levels were found in the same RNA sample of non-transfected cells using random hexamers, a nonsense-tail primer and no primer for the reverse transcription (Table 4, columns 3, lines 6–8). The RT primer-independent RT was less efficient in the case of the reference gene *Hprt* with the Hprt_Ex12 PCR primer generating, from the same RNA sample, PCR products at nearly undetectable levels using no RT primer (ct = 28.2; Table 4, column 2, line 8), low levels using the nonsense-tail RT primer (ct = 25,7; Table 4, column 2, line 7), and a high (total) level using random hexamers (ct = 21.4; Table 4, column 2, line 6). Such a variation of the efficiency in RT primer-independent reverse transcription between the different genes has already been described [18].

(4) The amount of transfected plasmid DNA strongly exceeded the transcription capacity, since the actual amount of plasmid-derived mRNA could have been synthesized already from 7% of the residual contaminating plasmid DNA (Table 4, line 2, columns 1 and 3, respectively). This was calculated by comparing the PCR product level derived from the RNA from *Pex11β* vector-transfected cells after nonsense-tail RT-qPCR using the 11b_Tail PCR primers (ct = 13.2; detecting only plasmid cDNA) and the 11b_Ex34 PCR primers (ct = 9.3; detecting plasmid and genomic cDNA) based on Equation (2). Thus, the transcription machinery, rather than the transfection rate, might be a limiting factor for the successful overexpression of a protein. This knowledge will help to minimize the amount of plasmid DNA used for transfection to a level below the maximal capacity of the transcription machinery and/or nuclear transportation [19,20,21,22]. This is important, as excessive amounts of DNA lead to the formation of high-molecular-weight DNA concatemers [22], which may affect the transcription of genomic DNA and cellular metabolism. Indeed, exogenous plasmid DNA induced a stress response and DNA damage in human hepatoma cells [23] and increased the number of neutrophils, when applied to the lung for gene therapy [24]. It is thought that CpG motifs present inside plasmids caused the stimulation of the immune system [25]. In addition, due to the limitation of the translation machinery, considerable amounts of plasmid-derived mRNA often remained untranslated [26,27].

### 3.3. High Amounts of Intracellular Plasmid DNA Disturbed the Removal of (Plasmid and Genomic) DNA during Standard RNA Isolation

Finally, we thought to check whether high intracellular levels of plasmid DNA disturbed genomic DNA removal during standard RNA isolation. The isolated RNA template, without reverse transcription, of non- and empty vector-transfected cells was analyzed for residual genomic DNA using a PCR primer pair, which binds inside exon 4 and thus detects genomic DNA as well as cDNA (11b_Ex4, Table 1). Nearly undetectable levels (ct = 30) of residual genomic DNA were found in both DNase I treated- and untreated RNA samples of non-transfected cells (Table 5, lines 1, 2) demonstrating the efficacy of RNAzol^®^ RT reagent in isolating pure (no DNA contamination) and intact RNA [28,29]. In contrast, high levels of genomic DNA remained in the untreated RNA samples of empty vector-transfected cells (ct = 24.6, Table 5, line 6). This was avoided by prior DNase I treatment (ct = 29.8, Table 5, line 5).

### 3.4. Necessity for a Selective Detection of Plasmid cDNA

To our knowledge, only a few studies have investigated how to distinguish genomic and plasmid DNA from transcribed cDNAs in RT-qPCR analysis of cells transfected with overexpression plasmids. Differentiation between the levels of plasmid and genomic DNA-derived mRNAs is facilitated when the DNA sequence inserted into the plasmid differs from the one from the genomic DNA. Since cDNA clones are usually inserted into expression vectors, one main differentiation criterion is the absence of introns within the insert sequence. Additionally, the plasmid DNA may contain either silent mutations [30] or synthetic or tagged introns [31,32]. These distinctions help-with the right primer-design-to discern between transfected plasmid and genomic DNA, provided that (i) there is no contaminating plasmid DNA, (ii) the splicing machinery can deal with extremely high amounts of precursor mRNA [33], and (iii) the intron tag does not modulate transcription. For most functional studies, however, introducing plasmid DNA containing cDNA fragments that are identical to the genomic DNA sequence is the method of choice for avoiding expression artefacts and mimicking the function of the endogenous protein during overexpression. When, however, the cDNA is inside the plasmid DNA, the genomic DNA-derived mRNA, and the plasmid-derived mRNA are identical, the distinction becomes crucial, as contaminations with genomic DNA or plasmid DNA cannot be excluded *a priori* [34]. Tagging plasmid-derived mRNA with a “unique” sequence during reverse transcription has already been proposed by Smith et al. [8] for avoiding the detection of pseudogenes, but as a novelty, the RT primer in our protocol contained a shorter poly(A) tail of 17 nt (thus binding to the plasmid DNA only) and, in addition, a complementary sequence of 12 nt, which again selectively annealed to the 3´-end of the plasmid DNA. This RT primer construct ensured that only a negligible amount of genomic DNA-derived mRNA (0.7%) was reverse transcribed bearing the nonsense tag.

In conclusion, we propose here a quantitative RT-PCR method for quickly detecting changes in plasmid-derived mRNA levels after transfection with overexpression plasmids, which can be helpful as guidelines in establishing and facilitating specific transfection protocols for different genes of interest.

## Figures and Tables

**Figure 1 mps-03-00040-f001:**
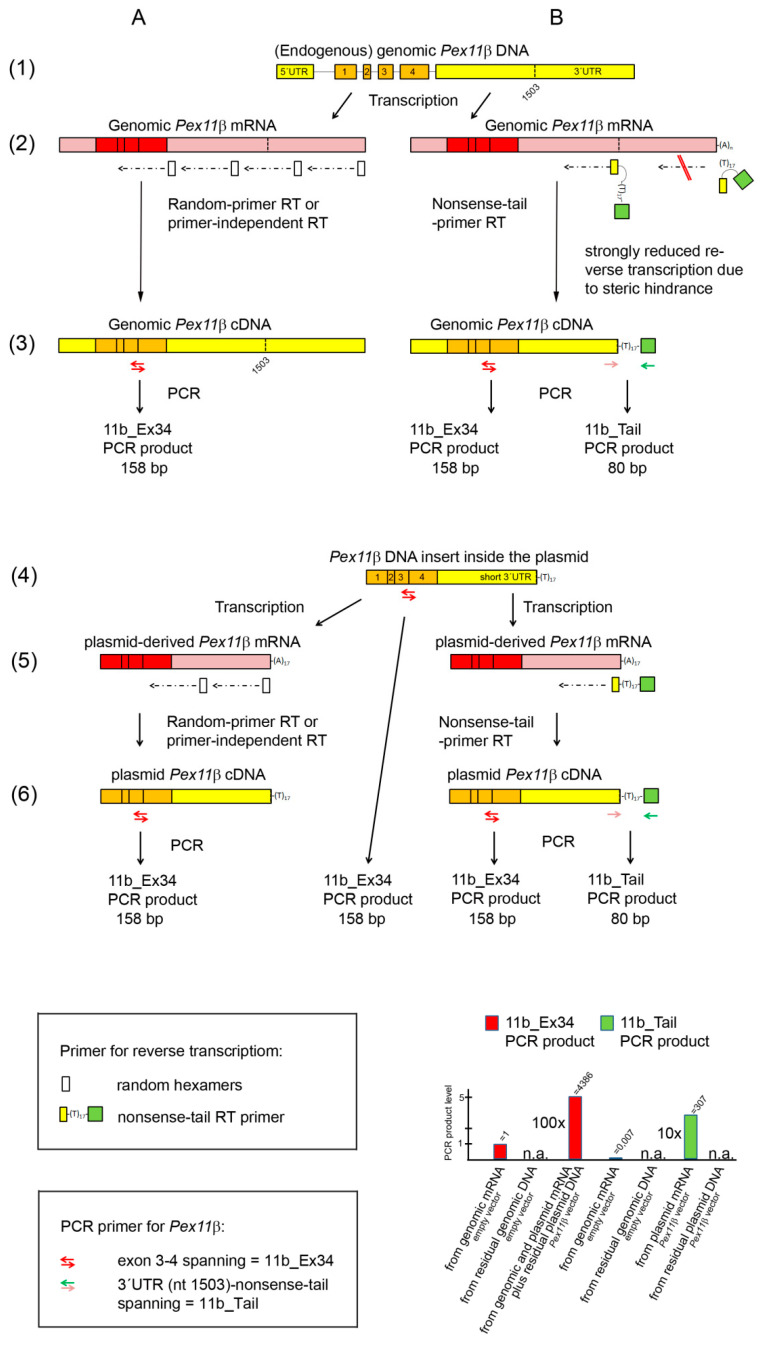
Schematic drawing of the mouse *Pex11β* gene (genomic DNA, line 1) and the *Pex11β* cDNA inserted into the expression vector (plasmid DNA, line 4), together with the corresponding transcripts (genomic mRNA in line 2, plasmid-derived mRNA in line 5) and different DNA copies (lines 3, 6) after reverse transcription using random hexamers (**A**) and the nonsense-tail RT primer (**B**), as well as and in the absence of an RT primer (no RT primer RT, A). Symbols for the RT primer and PCR primer are shown in the top and lower box on the right, respectively. Localizations of the three *Pex11β* PCR primer pairs (11b_Ex34, 11b_Tail) are shown. The histogram gives an overview of the relative PCR product levels obtained from genome- and plasmid-derived DNA and mRNA after random and nonsense-tail primer RT using the 11b_Ex34 (red) and 11b_Tail PCR primer (green).

**Table 1 mps-03-00040-t001:** **RT and PCR** primer sequences. List of primers used for the RT and qPCR reactions, including their gene names, gene bank accession (NCBI) numbers, sequences, binding positions, amplicon lengths, efficiency coefficients and their abbreviations used throughout this manuscript. The nonsense-tail sequence with no similarity to the mouse genome is highlighted in blue.

*Gene name* NCBI Accession Number	PCR Primer Sequence (5´-3´end)	Position Ms Transcript	Position Plasmid DNA	Am-pli-con size (bp)	Effi-cien-cy	Abbre-viation
*Pex11b*, NM_011069.3						
	Aaccgagccttgtactttgc aggcgaatctcataagcatca	532-551 689-669	281-300 418-438	158	1.99	11b_Ex34
	cgcctattgatggaacaagagact tccaggtcccacagtttctactc	685-708 780-758	434-457 507-529	96	1.99	11b_Ex4
	ctggtccttgcccacagagc ccgatcgcgatttcgataaaa	1470-1484 Ø	Ø 1721-1740	80	1.99	11b_Tail
*Hprt1* NM_013556.2						
	agtcccagcgtcgtgattag tttccaaatcctcggcataatga	159-178 246-224	Ø Ø	88	1.86	Hprt_Ex12

	RT primer sequence (5´-3´end)					
Nonsense-tail RT primer	Ggctagcgctaaagctattttttttttttttttt gagcaaactgat	1770-1782	1518-1530			

**Table 2 mps-03-00040-t002:** Detailed protocol of the reverse transcription and PCR reactions performed in this study. Variable steps are highlighted in gray.

High-capacity cDNA reverse transcription kit; Appl. Biosystems, Cat. No. 4368881
**Reaction**		Amount	Volume (µL)	defined as
**Reverse Transcription**	RNA template	2 µg	10.0	
+	10 x RT buffer		2.0	
+	25 x dNTP Mix	100 mM	0.8	
either	Multiscribe™ Reverse Transcriptase	50 U/µL	1.0	
or	Water, nuclease-free		1.0	no RT

either	10 x random hexamers		1.0	Random RT
or	Nonsense tail RT primer	1pmol/µL	1.0	Nonsense-tail RT
or	Water, nuclease-free		1.0	no RT primer RT

+	Water, nuclease-free		to 20.0	
10 min, 25°C; 120 min, 37°C; 5 min, 85 °C, ∞ 4°C

Maxima SYBR Green/Fluorescein qPCR Master Mix (2x); Thermo Fisher Scientific, #K0243
**Reaction**		Amount	Volume (µL)	Defined as
**PCR**	DNA template	cDNA from 0.2 µg total RNA	2.0	

+	Forward and reverse primer	2.5 pmol/µL	1.0 each	
	11b_Ex34			11b_Ex34 PCR
	11b_Ex4			11b_Ex4 PCR
	11b_Tail			11b_Tail PCR
	Hprt_Ex12			Hprt_Ex12 PCR

+	2 x SYBR Green qPCR Master Mix			
+	Water, nuclease-free		10.0	
10 min, 95°C; 40 cycles: 15 s, 95°C; 60 s, 60°C

**Table 3 mps-03-00040-t003:** After transfection with a *Pex11β* cDNA-containing expression vector, we measured extremely high increases in the level of the *Pex11β* PCR product using the exon 3–4 spanning primer (11b_Ex34, highlighted in gray). We hypothesized an amplification of the residual plasmid DNA, because DNase I treatment reduced the *Pex11β* PCR product level (column 4, lines 4 vs. 6) and genomic DNA will not be amplified with the 11b_Ex34 PCR primer. Cells were either treated with the transfection reagent only (no transfection), or transfected with an empty vector (empty vector) or a *Pex11β* cDNA expression vector (*Pex11β* vector) for 6 h and 12 h. Except for cells transfected with the *Pex11β* vector for 12 h, all RNA samples were subjected to random RT and PCR, without further DNase I treatment. In cells transfected with the *Pex11β* vector for 12 h, the same RNA was subdivided and either underwent no RT plus no DNase I treatment (line 5), or random RT, with (line 6) or without (line 4) a prior DNase I treatment. Ct values of the *Pex11β* and *Hprt* PCR products are given in columns 1 and 2, respectively. Data on the *Pex11β* PCR product were normalized to *Hprt* as the reference gene (column 3) and then related to non-transfected cells (calculation based on Equation (1)); the normalized PCR product level of non-transfected cells was set to 1 (column 4).

		1	2	3	4
	Experimental Condition	Primer *Pex11β*	Primer *Hprt*		
		11b_Ex34	Hprt_Ex12	normalized to *Hprt*	set to1
		*ct* values	*ct* values		
1	6 h, empty vector	17.10	19.30	1.13	1.00
2	6 h, *Pex11β* vector	9.00	20.55	674.87	595.98
3	12 h, empty vector	22.80	25.80	1.23	1.00
4	12 h, *Pex11β* vector	12.85	28.20	5395.47	4386.11
5	12 h, *Pex11β* vector, no RT	18.38	N/A		
6	12 h, *Pex11β* vector, DNase	14.85	26.50	469.68	381.81

**Table 4 mps-03-00040-t004:** We selectively reverse transcribed plasmid-derived mRNA using a nonsense-tail RT primer (nonsense-tail RT), which was quantified by PCR using a PCR primer pair (11b_Tail), matching to the 3’end of *Pex11β* transcript (forward primer) and to the nonsense-tail region (reverse primer). A comparison of the ct values of the PCR products of 11b_Tail and the *Pex11β* exon 3–4 spanning primer (11b_Ex34) revealed that only 7% (line 2, column 1 vs. 3, calculation based on Equation (2)) of the residual plasmid DNA was transcribed into (plasmid-derived) mRNA. The mouse endogenous genome-derived mRNA was reverse transcribed independently of an RT primer and thus in the nonsense-tail RT as well. However, only 0.7% of the total genomic DNA-derived mRNA was detected (calculation based on Equation (2)) using the 11b_Tail PCR primer, as calculated by comparing the ct values of the products of the 11b_Tail (column 1, line 1) and those of the 11b_Ex34 PCR primer (column 1, line 3).

		1	2	3
	Experimental Condition	Primer *Pex11β*	Primer *Hprt*	Primer *Pex11β*
		11b_Tail	Hprt_Ex12	11b_Ex34
	**6 h, Nonsense-Tail RT**			
1	empty vector	24.60	26.10	17.45
2	*Pex11β* vector	13.20	25.90	9.30
	**6 h, Random RT**			
3	empty vector	N/A	22.65	18.50
4	*Pex11β* vector	N/A	22.90	11.30
5	**6 h, No Transfection**			
6	random RT	N/A	21.44	21.02
7	nonsense-tail RT	26.83	25.7	20.17
8	no RT-primer RT	N/A	28.2	20.74
9	no RT	N/A	N/A	N/A

**Table 5 mps-03-00040-t005:** High amounts of plasmid DNA lower the purity of the RNA during standard isolation with RNazol^®^ RT. MC3T3-E1 cells were transfected with a transfection reagent only (no transfection) or an empty vector (empty vector). Without RT, the *Pex11β* exon primer (11b_Ex4) detected only contaminating genomic *Pex11β* DNA. RNA samples of empty vector-transfected cells (compare the ct values in lines 5 vs. 6), but not of non-transfected cells (compare the ct values in line 1 vs. 2) were contaminated with residual DNA.

	Experimental Condition	Primer *Pex11β*
	6 h, No Transfection	11b_Ex4
1	no RT, DNase	30.20
2	no RT, no DNase	30.50
3	random RT, DNase	21.50
4	random RT, no DNase	22.00
	**6 h, Empty Vector**	
5	no RT, DNase	29.80
6	no RT, no DNase	24.60
7	random RT, DNase	25.30
8	random RT, no DNase	24.50

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
