# Peer review of "Analysis of the Level of Plasmid-Derived mRNA in the Presence of Residual Plasmid DNA by Two-Step Quantitative RT-PCR"

_mps, 2020, doi:10.3390/mps3020040_

Round 1

Reviewer 1 Report

Barbara et al showed the interesting concept. However, extensive english editing is required for clarity to understand the readers. 

Also, some appropriate reference must be needed to show necessity and  importance of your work. 

Author Response

Answer to the Reviewers:

First of all, we would like to thank all reviewers for their helpful comments and criticism. We have dealt with each critical point of each reviewer, which improved the quality of our manuscript. All critical points of the reviewers (font: liberation; in black) followed by our response (font: arial) highlighted in different colors for each reviewer are found in the text below. All changes within the revised version of our manuscript are described in our answers for the reviewers and are highlighted in different colors for each reviewer in the revised version of our manuscript.

Reviewer 1: green color

Reviewer 1: Open Review

English language and style

(x) Extensive editing of English language and style required
( ) Moderate English changes required
( ) English language and style are fine/minor spell check required
( ) I don't feel qualified to judge about the English language and style

Yes

Can be improved

Must be improved

Not applicable

( )

(x)

( )

( )

( )

(x)

( )

( )

(x)

( )

( )

( )

(x)

( )

( )

( )

( )

(x)

( )

( )

Comments and Suggestions for Authors

Barbara et al showed the interesting concept. However, extensive english editing is required for clarity to understand the readers. 

Answer to Reviewer 1: The manuscript has been sent to the Regular MPIs Editing Service for English language and style (checking grammar and phrasing) and these corrections are all included in the revised version of the manuscript.

Also, some appropriate reference must be needed to show necessity and  importance of your work. 

Answer to Reviewer 1: We agree with the reviewer in this regard and included publications dealing with the problem of DNA contamination in RNA samples (Introduction, page 3, first paragraph). These references can be found in the beginning of the reference list (references 1-4).

Submission Date

19 February 2020

Date of this review

09 Mar 2020 04:01:51

Reviewer 2 Report

In this study, the authors proposed a simple way with minimal change to standard RT-PCR techniques toe distinguish between plasmid-derived mRNA, endogenous genome-derived mRNA and plasmid DNA by performing a plasmid mRNA-specific reverse transcription and the plasmid cDNA was additionally tagged with a nonsense tail. Using this method, it is easy to specifically determine the amount of mRNA derived from the transfected plasmid DNA in comparision to the endogenous genome-derived mRNA and thus the transfection efficiency. In general, this a well designed and perfomed study. And, the topic of this paper is interesting for the relative researchers.

Author Response

Answer to the Reviewers:

First of all, we would like to thank all reviewers for their helpful comments and criticism. We have dealt with each critical point of each reviewer, which improved the quality of our manuscript. All critical points of the reviewers (font: liberation; in black) followed by our response (font: arial) highlighted in different colors for each reviewer are found in the text below. All changes within the revised version of our manuscript are described in our answers for the reviewers and are highlighted in different colors for each reviewer in the revised version of our manuscript.

Reviewer 2: Grey Color, no Color in the revised version of the manuscript

Reviewer 2: Open Review

English language and style

( ) Extensive editing of English language and style required
( ) Moderate English changes required
(x) English language and style are fine/minor spell check required
( ) I don't feel qualified to judge about the English language and style

Yes

Can be improved

Must be improved

Not applicable

(x)

( )

( )

( )

(x)

( )

( )

( )

(x)

( )

( )

( )

(x)

( )

( )

( )

(x)

( )

( )

( )

Answer to Reviewer 2: The manuscript had been sent to the Regular MPIs Editing Service for English language and style (checking grammar and phrasing) and these corrections are all included in the revised version of the manuscript.

Comments and Suggestions for Authors

In this study, the authors proposed a simple way with minimal change to standard RT-PCR techniques toe distinguish between plasmid-derived mRNA, endogenous genome-derived mRNA and plasmid DNA by performing a plasmid mRNA-specific reverse transcription and the plasmid cDNA was additionally tagged with a nonsense tail. Using this method, it is easy to specifically determine the amount of mRNA derived from the transfected plasmid DNA in comparision to the endogenous genome-derived mRNA and thus the transfection efficiency. In general, this a well designed and perfomed study. And, the topic of this paper is interesting for the relative researchers.

Submission Date

19 February 2020

Date of this review

04 Mar 2020 06:39:46

Reviewer 3 Report

Comments

              I hope that the following comments would help the authors.

              The authors established the method to measure the amount of the mRNA transcribed from the plasmid DNA in cells which had been transfected with that plasmid. The paper seems interesting. However, it is very hard to understand the method because the explanation of this method is not clear. Especially, when the authors explain the method, they mention only the Figure 1. It is very difficult to find out which figure in Figure1 the authors point out in the sentence in the paper. Everything is only saying Figure1. there are many figures in the Figure1. Furthermore, the explanation for each figure in Figure1 is not detail. These things hinder me from understanding the methods and the results, and from taking away my interst in reading this paper.

  1. Please explain each figure in Figure1.
  2. Please show the method with figure (for example. with colored bars, etc.) step by step; what kinds of genes are amplified after RT and after PCR.
  3. Please show the primers (RT primer, forward primer, reverse primer) in each figure in Figure1.
  4. Please link the results (for example, Table 3, etc.) to the figure in Figure1 to understand the results and reaction-conditions.
  5. Please show the results of the DNA-bands in the agarose gel after the RT-qPCR to know the lengths of the RT-PCR products. This helps the understanding the methods.

Minor comments;

Please explain the meanings of ‘’pDNA, gmRNA, and pmRNA’’ in the last line in the Figure1.

Author Response

Answer to the Reviewers:

First of all, we would like to thank all reviewers for their helpful comments and criticism. We have dealt with each critical point of each reviewer, which improved the quality of our manuscript. All critical points of the reviewers (font: liberation; in black) followed by our response (font: arial) highlighted in different colors for each reviewer are found in the text below. All changes within the revised version of our manuscript are described in our answers for the reviewers and are highlighted in different colors for each reviewer in the revised version of our manuscript.

Reviewer 3: yellow color

Reviewer 3: Open Review

English language and style

( ) Extensive editing of English language and style required
(x) Moderate English changes required
( ) English language and style are fine/minor spell check required
( ) I don't feel qualified to judge about the English language and style

Yes

Can be improved

Must be improved

Not applicable

( )

(x)

( )

( )

( )

(x)

( )

( )

( )

( )

(x)

( )

( )

( )

(x)

( )

( )

( )

(x)

( )

Answer to Reviewer 3: The manuscript had been sent to the Regular MPIs Editing Service for English language and style (checking grammar and phrasing) and these corrections are all included in the revised version of the manuscript.

Comments and Suggestions for Authors

Comments

              I hope that the following comments would help the authors.

              The authors established the method to measure the amount of the mRNA transcribed from the plasmid DNA in cells which had been transfected with that plasmid. The paper seems interesting. However, it is very hard to understand the method because the explanation of this method is not clear.

Answer to Reviewer 3: We agree with Reviewer 3 that we should explain in more detail (step-by-step) the principle of the method and we changed our manuscript according to his suggestions.

Especially, when the authors explain the method, they mention only the Figure 1. It is very difficult to find out which figure in Figure1 the authors point out in the sentence in the paper. Everything is only saying Figure1. there are many figures in the Figure1. Furthermore, the explanation for each figure in Figure1 is not detail. These things hinder me from understanding the methods and the results, and from taking away my interst in reading this paper.

  1. Please explain each figure in Figure1.

We agree with the Reviewer 3 that Figure 1 should be improved in quality and comprehensibility and changed it in this regard: we gave full names (no abbreviations) for all drawings, we defined lines 1-6 (left side) and columns (A and B) to define the position of all drawings (and reactions) within the manuscript text (e.g. line 2B).

  1. Please show the method with figure (for example. with colored bars, etc.) step by step; what kinds of genes are amplified after RT and after PCR.

In the new Figure 1, genomic DNA and subsequent RT (line 2, 5) and PCR (lines 3, 6) reactions were shown on the left site for random RT (column A) and on the right site for nonsense-tail RT (column B). Moreover, we added a histogram on the lower, right site on the relative PCR product levels of genome- and plasmid-derived DNAs and mRNAs using the 11b_Ex34 and 11b_Tail PCR primers.

  1. Please show the primers (RT primer, forward primer, reverse primer) in each figure in Figure1.

In the new Figure 1, we added a box on the upper, right site containing the symbols for the RT primers (upper box) and PCR primers (lower box). Binding sites of the RT primers are shown in the lines for reverse transcription (lines 2, 5) and of the PCR primers in the lines for PCR (lines 3, 6).

  1. Please link the results (for example, Table 3, etc.) to the figure in Figure1 to understand the results and reaction-conditions.

In the new version of Results and Discussion, data (i) were given in the text (ct values plus a link to the respective position inside the respective Tables), (ii) were highlighted in grey in the Tables and (iii) were linked to Figure 1 as proposed by the Reviewer 3.

For 1-4. All these changes were highlighted in yellow in the revised version of our manuscript

  1. Please show the results of the DNA-bands in the agarose gel after the RT-qPCR to know the lengths of the RT-PCR products. This helps the understanding the methods.

With regard to the demonstration of the PCR products in an agarose gel, we performed new experiments. We used Pex11β vector-transfected cells, which underwent either random hexamer RT or nonsense-tail primer RT. The PCR products are visualized in a 3% agarose gel - this was necessary because of the small size of the qPCR amplicons. This agarose gel wis shown in the new Supplementary Figure 1.

Supplementary Figure 1: Agarose gel electrophoresis (3%) of PCR products from the cDNA of Pex11b vector-transfected cells (6 h post-transfection). Total RNA was reversely transcribed using random hexamers (random RT), or the nonsense-tail RT primer (nonsense-tail RT). Lane 1: random RT, the Hprt_Ex12 PCR primer product of 88 bp; Lane 2: random RT, the 11b_Ex34 PCR primer product of 158 bp; Lane 3: nonsense-tail RT, the 11b_Tail PCR primer product of 80 bp; Lane 4: random RT, 1the 1b_Ex4 PCR primer product of 96 bp. The 100-bp DNA molecular weight marker is shown on the left.

Please find the agarose gel in the attached file.

Minor comments;

Please explain the meanings of ‘’pDNA, gmRNA, and pmRNA’’ in the last line in the Figure1

We did not use abbreviations any more in the new Figure 1.

Submission Date

19 February 2020

Date of this review

02 Apr 2020 07:30:23

Reviewer 4 Report

In this study, the authors have described the quantitative RT-PCR method to quickly detect changes in plasmid-derived mRNA levels after transfection with overexpression plasmids. This technique enables to distinguish between plasmid-derived mRNA, endogenous genome-derived mRNA and plasmid DNA. This manuscript was a pleasure to read. It is clear, well-written, and accurately describes the hypothesis tested and the methodologies used. The study design is logical and the experimental results are accurately described without bias. I only have one minor comment. I wish every manuscript that I reviewed were this well written.

Minor point: In section 2: Materials and Methods, subsection 2.1: Materials (page 2 of the manuscript) and in table 1 (page 3 of the manuscript): the efficiency coefficient is not defined. It would be valuable if the Authors could present the mathematical formula used for the efficiency coefficients calculations.

Author Response

Answer to the Reviewers:

First of all, we would like to thank all reviewers for their helpful comments and criticism. We have dealt with each critical point of each reviewer, which improved the quality of our manuscript. All critical points of the reviewers (font: liberation; in black) followed by our response (font: arial) highlighted in different colors for each reviewer are found in the text below. All changes within the revised version of our manuscript are described in our answers for the reviewers and are highlighted in different colors for each reviewer in the revised version of our manuscript.

Reviewer 4: blue color

Open Review

English language and style

( ) Extensive editing of English language and style required
( ) Moderate English changes required
(x) English language and style are fine/minor spell check required
( ) I don't feel qualified to judge about the English language and style

Yes

Can be improved

Must be improved

Not applicable

(x)

( )

( )

( )

(x)

( )

( )

( )

(x)

( )

( )

( )

(x)

( )

( )

( )

(x)

( )

( )

( )

Answer to Reviewer 4: The manuscript had been sent to the Regular MPIs Editing Service for English language and style (checking grammar and phrasing) and these corrections are all included in the revised version of the manuscript.

Comments and Suggestions for Authors

In this study, the authors have described the quantitative RT-PCR method to quickly detect changes in plasmid-derived mRNA levels after transfection with overexpression plasmids. This technique enables to distinguish between plasmid-derived mRNA, endogenous genome-derived mRNA and plasmid DNA. This manuscript was a pleasure to read. It is clear, well-written, and accurately describes the hypothesis tested and the methodologies used. The study design is logical and the experimental results are accurately described without bias. I only have one minor comment. I wish every manuscript that I reviewed were this well written.

Minor point: In section 2: Materials and Methods, subsection 2.1: Materials (page 2 of the manuscript) and in table 1 (page 3 of the manuscript): the efficiency coefficient is not defined. It would be valuable if the Authors could present the mathematical formula used for the efficiency coefficients calculations.

Answer to Reviewer 4: As recommended by Reviewer 4, we explain in more detail the calculations of the efficiency coefficients and of the comparison of different PCR product levels by giving the mathematical formula in the new version of our manuscript. These changes were included in the chapter Materials and Methods, 2.3. RNA isolation, RT reaction and quantitative PCR analysis, (pages 9-10), the respective text is highlighted in blue.

Submission Date

19 February 2020

Date of this review

25 Mar 2020 19:35:32

Round 2

Reviewer 3 Report

Dear Authors,

Thank you for answering all my comments, especially the new Figure 1 is easier to understand the procedures. I appreciate your revise.

I still have some minor comments. Please consider these comments.

  1. It seems no red line in Table 2 which is mentioned in the legend.
  2. I think that it is better to write ‘’ set to 1’’ (add ‘to’) instead of ‘’set 1’’ at the top of column 4 in Table 3.
  3. I think that it is ‘’column 4, lines 4 vs. 6’’ instead of ‘’column 4, lines 5 vs. 6’’ at line 4 from the top in the legend of Table 3.
  4. Please see the sentence; Indeed, in the absence of RT, no PCR product was found with 11b_Ex34 PCR primer (Table 3, column 1, line5…’’ But there is number, 18.38 in Table 3 at lines 5-6, page 7.
  5. Please set each Number (1), (2), etc. and each figure to the same horizontal line in Figure 1.

Author Response

Comments to the Reviewer:

We would like to thank the Reviewer for his careful proofreading. All minor critical points raised by the reviewer were changed according to his suggestions. In addition, we removed the highlightening of the text in different colors for each reviewers of the first revised version of our manuscript.

Thank you for answering all my comments, especially the new Figure 1 is easier to understand the procedures. I appreciate your revise.

I still have some minor comments. Please consider these comments.

  1. It seems no red line in Table 2 which is mentioned in the legend.

The reviewer is correct that there is no red line in Table 2. The legend of Table 2 was changed accordingly (page 9, last line from the bottom).

  1. I think that it is better to write ‘’ set to 1’’ (add ‘to’) instead of ‘’set 1’’ at the top of column 4 in Table 3.

As suggested by the reviewer, we changed the text „set to 1“ instead of „set 1“ in the heading of Table 3 and in the legend of Table 3 (page 13, line 3).

  1. I think that it is ‘’column 4, lines 4 vs. 6’’ instead of ‘’column 4, lines 5 vs. 6’’ at line 4 from the top in the legend of Table 3.

The reviewer is correct: it is column 4, lines 4 vs. 6 instead of column 4, lines 5 vs. 6. We changed the text in Table 3 accordingly.

  1. Please see the sentence; Indeed, in the absence of RT, no PCR product was found with 11b_Ex34 PCR primer (Table 3, column 1, line5…’’ But there is number, 18.38 in Table 3 at lines 5-6, page 7.

The reviewer is correct - we made a mistake. To demonstrate that there is - in the absence of RT - no PCR product (too long) from genomic Pex11b DNA using the 11b_EX34 PCR primer, we had to use non-transfected or empty vector-transfected cells. This is shown in Table 4, column 3, line 9. We changed the text accordingly (page 13, line 10).

In Table 3, column 1, line 5, the ct value of 18.38 in Pex11b DNA vector-transfected cells represents the level of plasmid Pex11b DNA.

That there is - in the absence of RT - no PCR product (too long) from genomic Hprt DNA using the Hprt_EX12 PCR primer is additionally shown in Table 3, column 2, line 5 (and not only shown in Table 4, column 2, line 9) – this fact was added on page 13, line 12).

  1. Please set each Number (1), (2), etc. and each figure to the same horizontal line in Figure 1.

As suggested by the reviewer, we set the numbers (1) (2) in Figure 1 to the same horizontal line as the drawings of the genomic and plasmid DNAs, mRNAs and cDNAs.
